# Analysis of the Frequency and Correlated Factors of Midpalatal Suture Maturation Stages in Young Adults, Based on Cone Beam Computed Tomography Imaging

**DOI:** 10.3390/jcm11236959

**Published:** 2022-11-25

**Authors:** Juan Carlos Silva-Montero, Ignacio Faus-Matoses, David Ribas-Pérez, Hourieh Pourhamid, Beatriz Solano-Mendoza

**Affiliations:** 1Department of Stomatology, Faculty of Dentistry, University of Seville, 41009 Seville, Spain; 2Department of Stomatology, Faculty of Dentistry, University of Valencia, 46010 Valencia, Spain

**Keywords:** midpalatal, suture, stages, adults, frequency, CBCT

## Abstract

Background: The choice of whether to perform a palatal disjunction in constricted maxilla has traditionally been decided based on the age of the patients, although there are gradually increasing references to the fact that this is not a determining factor. The main goal of this study was to evaluate the frequency of the different stages of midpalatal suture maturation in a sample of young adults between 15 and 30 years of age. Other objectives also included analyzing the possible correlation the maturation stages could maintain with sex and age groups. Methods: 142 Cone Beam Computed Tomography (CBCT) scans of young adults were performed. The images were divided into four age groups based on age ranges of 15–18, 19–22, 23–26, and 27–30 years. Each group consisted of 26, 41, 39, and 36 patients, respectively, which were classified using Angelieri’s method. In addition, sex and age groups were considered as variables, and the possible correlation of the prevalence of each one, according to age and sex, was studied. Results: the sample was classified into 4.9% stage B; 52.1% stage C; 27.5% stage D; and 15.5% stage E. In addition, no statistically significant correlation between sex and the maturation stages was found, but more advanced stages did appear to be related to the chronological age of the subjects. Conclusions: The frequency of maturational stages where the suture is shown to be consolidated did not appear to be as high as expected; therefore, the idea of rejecting transverse plane treatment in a conventional manner in an out-of-growth patient should be discarded.

## 1. Introduction

Malocclusions in the transverse plane, especially those caused by a constricted maxilla, can lead to posterior crossbite; lack of space for dental material, which is manifested by dental crowding; functional alterations, which are reflected in the narrowing of the pathways and lead to alterations in tongue position and breathing, among others; V-shaped maxillary arch; and high and narrow vaulted palate [1].

The origin of this lack of maxillary transversal development may be due to a multitude of factors, among which the followings are the most important: the presence of habits [2], such as thumb sucking; obstruction of the upper airway, leading to obstructive sleep apnea; the presence of facial clefts [3]; non-syndromic palatal synostosis [4]; or the presence of certain syndromes, such as the Klippel-Feil syndrome [5], congenital stenosis of the nasal pyriform aperture, Marfan syndrome, craniosynostosis (Appert, Crouzon, Carpentier disease) and Treacher Collins syndrome [6].

Regarding the diagnosis of maxillary transverse underdevelopment, jaw deficiency can be assessed by different methods, such as clinical assessment by analysis of models or occlusograms, and radiological assessment, either by frontal radiographs or three-dimensional imaging techniques, such as cone beam computed tomography (CBCT), the latter providing a much more accurate assessment.

Through clinical evaluation, we can mainly detect a constricted maxilla by finding buccal corridors in smile assessment, the presence of posterior crossbite, asymmetries in the shape of the maxillary arch, the depth of the palatal vault, as well as determining the predominant respiratory pattern.

Three-dimensional imaging techniques, such as cone beam computed tomography (CBCT), have allowed us to accurately visualize the craniofacial region. Thus, it offers a 3D representation of the maxillofacial structures without the superimposition of contiguous anatomy and provides the possibility to perform sections on the axial plane of the apical bases at different levels, helping the practitioner to make an accurate diagnosis [7].

From a histological point of view, the midpalatal suture undergoes changes during an individual’s growth phases, which allows us to determine the moment at which we can perform a disjunction. At birth, the suture is broad and later acquires a squamous shape. During puberty (at the age of 15 to 16, the transverse suture in most cases was found to consist of a narrow sheet of connective tissue with inactive osteoblasts covering smooth bone edges), the suture becomes sinuous again, which favors its development. From this time onwards, its morphology changes until it is completely interdigitated by the processes. In addition, on the posterior margin of the hard palate, apposition was found to continue for several years until the age of 18 [8].

Once the lack of transversal development of the maxilla has been diagnosed, we must be aware of what type of treatment can be carried out in each case, as it will depend directly on the origin of the deficiency, as well as on age, given that, at an early age (9–13 years), the constriction can be solved by slow expansion. However, it is very common to resort to Rapid Maxillary Expansion (RME) in adolescents with compressed maxillary [9].

In the case of a skeletal origin, it is the bony base that does not have sufficient width due, in most cases, to the lack of tensile forces from the masticatory function, which are not sufficient to stimulate the bony apposition at the edges of the palatal suture which is a joint [10]. This means that re-establishing a correct functional pattern of the masticatory system, especially in young adults, is essential for a logical and consistent therapy that will stimulate the suture to produce physiological bone since the heavy forces caused by the RME destroys de vascularity of the suture zone and the bone separation is repaired by apposition of scar tissue-like, that suffers an unavoidable relapse after 6 months.

However, to be able to apply disjunction, it is necessary to know the state of maturation of the midpalatal suture and its degree of development, as this will determine whether rapid maxillary expansion (RME) can be performed.

The ossification of the suture is associated with the skeletal maturation of the individual, which, traditionally, was determined from a wrist x-ray, which helped us to determine the growth period of the individual, as described by Björk and Grave [11]. Subsequently, taking advantage of lateral radiography in orthodontic diagnosis, and in order to reduce the patient’s exposure to ionizing radiation, it was proposed to determine the degree of bone maturity according to the morphology of the cervical vertebrae [12]; but later on, the same authors suggested that this method is no longer considered the most reliable approach [13].

Nevertheless, this only gave an idea of the degree of biological maturation of an individual, as in no case did it confirm with certainty the degree of ossification of the palatal suture. In 2013, Angelieri et al. [14] proposed a method of individual assessment of midpalatal suture morphology and determined five stages of maturation (A, B, C, D, and E) as a way to provide more reliable clinical data, allowing to discern and classify the midpalatal suture according to the degree of the sutural line separation (Figure 1).

The main goal of this study was to evaluate the frequency of the different stages of midpalatal suture maturation in a sample of young adults between 15 and 30 years of age. Other objectives also included analyzing the possible correlation the maturation stages could maintain with sex and age groups.

## 2. Materials and Methods

### 2.1. Design and Sample

The study sample consisted of 142 CBCT scans of young adults divided into four age groups ranging from 15 to 18 years and 11 months, from 19 to 22 years and 11 months, from 23 to 26 years and 11 months, and from 27 to 30 years and 11 months. In addition, each consisted of 26 (15 females and 11 males), 41 (26 females and 15 males), 39 (30 females and 9 males), and 36 (20 females and 16 males) patients, respectively.

This retrospective analytical observational study did not need to be approved by any Ethical Committee since these CBCT images were already in the database of the Faculty of Dentistry of the University of Seville, where they had their corresponding radiological studies previously performed for other purposes, such as preparation for wisdom tooth extractions or orthodontic study of included canines.

The sample distribution showed 35.9% of male patients, with a total number of 51 subjects, while the female gender represented 64.1%, with 91 subjects. As can be seen, this study englobes a greater number of females due to the nature of systematic random sampling selection.

### 2.2. Inclusion and Exclusion Criteria

The inclusion criteria taken into account were being a patient at the Faculty of Dentistry of Seville; presenting a CBCT in the databases of the faculty, due to the aforementioned purposes; belonging to one of the age groups determined in the study design; that the data relating to sex were concise; and that the patient signed the informed consent form of the faculty of Dentistry in which they accept that their data can be used for educational and/or research purposes.

The characteristics that marked the exclusion criteria were those that did not present high-resolution CBCT images or those in which the midpalatal suture could not be fully observed; the presence of noise in the radiographic images; patients with previous orthodontic treatment; a history of maxillofacial trauma; the presence of a disjunctor; or the presence of cleft lip or palate.

### 2.3. Methodology

CBCT scans were selected from patients aged between 15 and 30 years of both sexes from January 2013 to January 2022. Groups younger than those indicated were not taken into account because they were not found in the faculty’s databases, as CBCTs are not performed for extractions of wisdom teeth or canines included at such young ages.

All CBCT scans were performed using the Planmeca Promax^®^ 3D Sirona scanner adjusted to the following specifications: a field of view of at least 8 cm, 90 kV, 12 mA, 1057 DAP (mGy/${\mathrm{cm}}^{2}$), a voxel size of 200 µm, and an exposure time of 12.25 s. The CBCT images were analyzed using Planmeca Romexis^®^.

The patient’s head was positioned in the three planes of space in a standardized protocol, instructing patients to maintain the natural position of the head and to occlude with their usual maximum intercuspation. Thus, the software cursor for image analysis was placed in the patient’s mid-sagittal plane for the axial and coronal view. For the sagittal view, the patient’s head was adjusted so that the horizontal reference line matched the midpalatal region (palatal plane), which is the annular bone between the superior and inferior cortexes (Figure 2).

For the evaluation of the sutural stage, the sections were selected according to the protocol described by Angelieri [14], although a particular difficulty was encountered in cases of very curved palates, where it was impossible to adjust the axial reference line to coincide with the palatal plane for correct assessment.

As reflected in the literature for cases with thick or curved palate [14,15,16], for greater precision when assessing the suture, the measurements were performed on different heights according to the palate shape for each part of the suture in anterior, middle, and posterior positions.

Subsequently, visualization and classification of the skeletal maturation stage of the midpalatal suture according to Angelieri’s method was performed on the CBCT axial slices themselves [14] Figure 3.

Training and calibration for accurate assessment were performed by an experienced and trained orthodontist (BSM) using 18 randomly selected CBCT slices of young adults (18–30 years) of both sexes. The observer (JCSM) was given a detailed explanation of the morphological characteristics of each stage of maturation, according to Angelieri [14].

Calibration was performed twice with a three-week washout period to avoid intra-examiner error and the recorded data were subjected to a concordance analysis to check for inter-examiner reliability.

The weighted kappa coefficients were calculated for the assessment of both intra- and inter-examiner measurement error using an online kappa index calculator. This site was created and commissioned by Justus Randolph’s original Java version, designed and programmed by Walubengo M. Singoro (2008). HTML and JavaScript port programmed by Alexander Cole (2016) [17].

### 2.4. Statistical Analysis

All statistical processing was performed using SPSS version 28 software (SPSS Inc., Armonk, NY, USA) for iOS. The chi-square test was used to determine the relationship between the different stages of maturation of the midpalatal suture and age, as well as the correlation with the sex of the patient.

A binary logistic regression model was performed using the maturation stage as the dependent variable. The independent variables were age and sex. The codes were 1 for males and 2 for females; and in the case of age, 4 groups were created, presenting 1 for the 15–18 age group; 2 for the 19–22 age group; 3 for the 23–26 age group; and 4 for the 27–30 age group.

The impact of each factor on the dependent variable used was expressed as OR with its 95% confidence interval (95% CI). Statistical significance for all statistical tests was set at *p* < 0.05.

## 3. Results

As for the distribution of the sample, 35.9% were male, while 64.1% were female. In terms of the suture maturation classification, 4.9% were classified as stage B; 52.1% as stage C; 27.5% as stage D; and 15.5% as stage E.

According to the age groups into which the sample was divided, there was a statistically significant association between age and maturation of the midpalatal suture (*p* < 0.01) (Table 1). On the other hand, no statistically significant correlation (*p* = 0.423) between sex and degree of suture ossification was found (Table 2).

Regarding the different stages, only the age group between 15–18 years included cases with stage B suture, and no cases with stage A suture were detected (Table 3).

## 4. Discussion

The key success of any treatment is the accurate diagnosis of the etiopathogenesis of the problem. Therefore, the treatment options can be logical and coherent and can ensure stability and correct function in the future. Having that in mind, in cases where a physiological therapy able to correct the malocclusion together with the function of the masticatory system is possible, it should be the clinician’s ideal option. The literature supports the fact that breaking the suture means taking a little immediate advantage of less than half of the expansion realized by the screw (<4.5 cm) [18] with a great unavoidable relapse after 6 months and much more over time, especially in growing subjects in which the suture will disappear, and no more bone will be added. It is also proven that in adults, breaking the suture means losing its viscoelasticity forever [19].

That being said, there are also many authors who defend RME as an ideal, and even in some cases, the gold standard and only option to treat a constricted maxilla in a non-invasive way [20,21]. Despite the resounding success of the RME protocol in daily clinical practice, there is still no consensus protocol on the age limit for disjunction. This is mainly due to the large possibility of different maturational stages of the midpalatal suture in patients of the same chronological age. The uncertainty of suture ossification creates uncertainty in treatment choices since applying heavy forces, such as ones in RME on obliterated suture, may have consequent risks of important side effects such as molar inclination, periodontal damage, root and bone resorption, among others [22].

Given the midpalatal suture variations in the maturation stages among different age groups, there is no solid rule that RME cannot be applied at specific age times. In 2022, a study by Rabah et al. [9] compared slow maxillary expansion with rapid palatal expansion in patients aged 12–15 years, and found that even with slow expansion protocols, changes in the suture can occur, leading to the disjunction of the suture itself, so that at ages as young as 15 years, it is possible to find the suture completely open, and it is possible to solve the transverse problem without inducing unnecessary morbidity in the patient.

The authors themselves also published an article comparing RME with slow palatal expansion in early adolescence (between the ages of 12–16 years), where they found that there was no significant difference between the two groups, and in this age group, it can be a good alternative to RME [23].

However, we can also find patients in adulthood presenting with an open suture, usually stage C, who are candidates to be treated by Mini-screw Assisted Rapid Palatal Expansion (MARPE). However, undergoing surgical procedures, such as Surgically Assisted Rapid Palatal Expansion (SARPE), leads to an unnecessary increase in morbidity, a decrease in patient comfort, and an increase in patient costs [24].

For this reason, a diagnostic method in which it is possible to assess the maturation stage of the midpalatal suture itself with certainty, safety, and reliability is of great importance. So far, the method of *Angelieri* [14] is considered the gold standard since, through CBCT images, it avoids overlapping structures and has the aforementioned characteristics. The sample was classified into 4 age groups, comprising a first group of 5–10 years, another of 11–13 years, another of 14–18 years, and the last of over 18 years. According to the results, the fusion of the midpalatal suture is directly related to age; and the female sex shows a tendency to mature more prematurely than the male sex. In this case, only 16% of adults (over 18 years of age) had an open midpalatal suture, most of them belonging to the C-stage group.

The author later expanded this study [25], where CBCT images of 78 subjects (64 of them female versus only 14 male) with an age range of 18–66 years were analyzed. It was determined that the midpalatal suture was not fused in 12% of the participants. Therefore, it concluded that sex and chronological age were not significant predictors of midpalatal suture maturation stages. Therefore, the need for individual assessment to determine the stage of midpalatal suture maturation was justified to make the clinical decision between rapid maxillary expansion and surgically assisted rapid maxillary expansion.

In 2017, Tonello et al. [15] used the same methodology as that described by *Angelieri* [14], but in this study, the sample was restricted to the age range of 11–15 years, in which a good response to RME is generally expected. These authors concluded that the suture remained open and, therefore, susceptible to disjunction (stages A, B, and C) in 76.2% of the adolescents. This result shows the reliability of the method due to the fact that, in this age group, RME is usually performed in a more systematic and successful way.

Regarding gender distribution, a higher prevalence of stage C was found in females than in males (56.8% and 42.5%, respectively), which is the opposite of the results found in the present study (53.3% and 72.7%, respectively), taking into account that the closest age range to the one assessed by Tonello et al. [15] was 15–18 years. However, even though the male group was found to predominate, the difference was not statistically significant.

Ladewig et al. [16] developed a study based on the same methodology, but this time the age of the sample was restricted to young subjects between 16 and 20 years in order to justify the request for CBCT in patients with a doubtful prognosis for conventional RME. All maturation stages were observed, with 52.6% of the sample presenting an open suture (stages A, B, and C), which gives information that even in more advanced age groups, disjunction could be successful without having to refer the patient for surgical intervention. Furthermore, no statistically significant differences were observed in the female sex, where the suture ossified earlier compared to the male group.

In comparison with this study, in the age range of 15–18 years, the suture was found to be open in 88.4% of cases (Stages A, B, and C), whereas, in the age range of 19–22 years, it was observed in 68.3%, which agrees with the results obtained in the previous article cited. Furthermore, in this article, no statistically significant differences were found in the female sex maturing earlier than the male sex.

Jiménez-Valdivia et al. [26] published a study in 2019 where they evaluated the midpalatal suture of 200 people, grouped into 3 age intervals: one being 10–12 years, another 16–20, and the last 21–25 years. These were not evenly divided, 100 of them represent the older age group, so comparisons with the present study cannot be reliable.

Jiménez-Valdivia et al. [26] came to the conclusion that in the second group (16–20 years), 21.2% of the patients presented an open suture, with no statistically significant differences between sexes, as in this study, whereas in the closest age group (19–22 years), the suture was found open in 68.3% of the cases. In the older age group (21–25 years), the results were that 17% of the participants still had the midpalatal suture with the possibility of disjunction without surgical assistance, yet in the present sample, the closest age group (23–26 years), 53.8% of the subjects still had the suture open. 

On the other hand, the study of Reis et al. [27] analyzed the CBCT images of 289 women and 198 men aged between 15 and 40 years, dividing the sample into five groups, which will be broken down as follows: in the first group (15–20 years), 45.7% of the individuals still had an open midpalatal suture, all of them belonging to stage C. In the second group (21–25 years), the stage decreased to 34.1%, and in the third group (26–30 years), 29.4%, therefore susceptible to disjunction without surgical assistance.

The two remaining groups mentioned above (31–35 years and 36–40 years) showed that the suture remained unconsolidated in 22.8% and 32.5%, respectively.

In comparing the results of the present study, it can be concluded that 61.5% of those belonging to the 15–18 age group are at stage C. In the age group 23–26 years, 53.8% of the sample belonged to stage C of maturation; and in the age group 27–30 years, 22.2% were at stage C.

Extrapolating the results with the whole study sample demonstrates the absolute indication of CBCT scans as a key requirement for a correct diagnosis and treatment planning of transverse plane problems.

## 5. Conclusions

-The frequency of maturation stages where the suture consolidates did not seem to be as high as expected, remaining open in 88.4%, 68.3%, 53.8%, and 22.2% in the age groups 15–18 years, 19–22 years, 23–26 years and 27–30 years of age, respectively. Therefore, the idea of rejecting transverse plane treatment in a conventional manner in a non-growing patient should be discarded.-CBCT is essential to determine the possibility of palatal disjunction, especially in postadolescents and young adults.-Chronological age is directly proportional to the degree of maturation of the midpalatal suture, but it was found that even in the oldest age group, 22.2% of patients still have an open suture.-According to this study, sex does not seem to have a statistically significant influence on the maturation stage of the midpalatal suture.

## Figures and Tables

**Figure 1 jcm-11-06959-f001:**
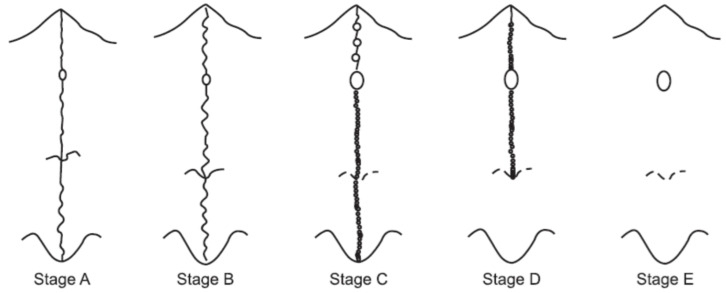
Midpalatal suture morphology classification according to Angelieri et al. [14].

**Figure 2 jcm-11-06959-f002:**
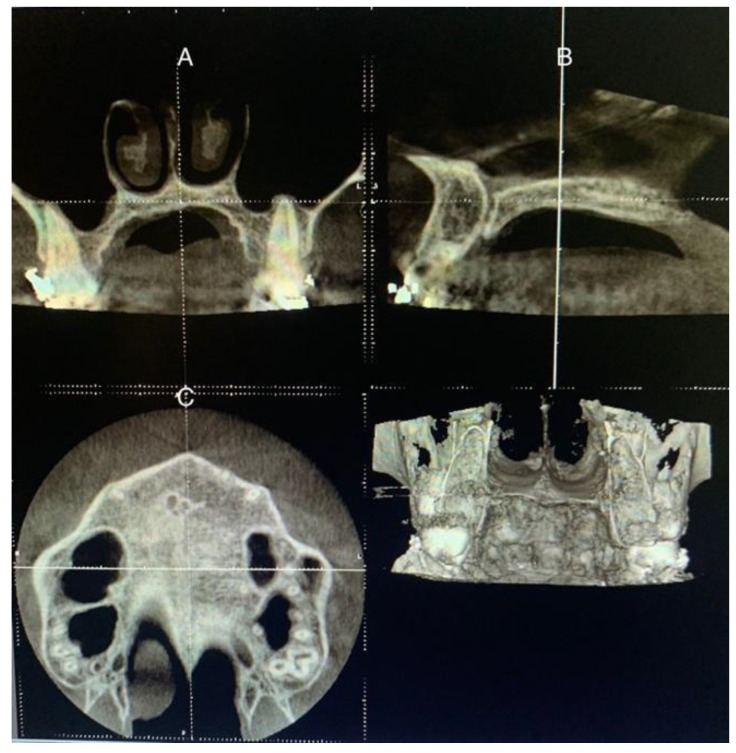
Procedures on CBCT sections to determine the stage of maturation of the palatal suture. (**A**) Coronal view. (**B**) Sagittal view, the reference line placed on the hard palate. (**C**) Axial view.

**Figure 3 jcm-11-06959-f003:**
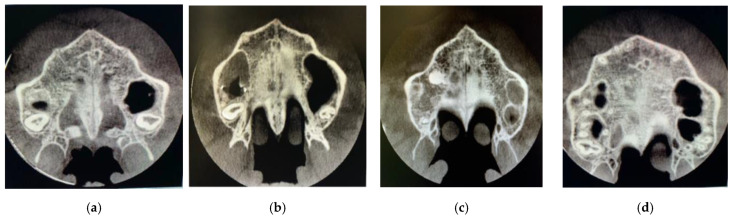
Axial slices of the different stages of the suture in the study. Stage B (**a**), stage C (**b**), stage D (**c**) and stage E (**d**).

**Table 1 jcm-11-06959-t001:** This table shows the statistical association between the variable age and suture maturation.

Symmetrical Measurements			
		Value	Approximate significance
Nominal by nominal	Phi	0.649	<0.01
	V de Cramer	0.375	<0.01
Number of valid cases		142	

**Table 2 jcm-11-06959-t002:** This table shows the statistical association between the variable sex and suture maturation.

Chi-Square Tests			
	Valor	Gl	Asymptotic significance (bi-lateral)
Pearson’s chi-square	3.876	4	0.423
Plausibility ratio	4.277	4	0.370
Number of valid cases	142		

**Table 3 jcm-11-06959-t003:** Results of the study.

Edad		Etapa A		Etapa B		Etapa C		Etapa D		Etapa E	
		N	%	N	%	N	%	N	%	N	%
15–18	F	0	0	5	33.3	8	53.3	2	13.3	0	0
	M	0	0	2	18.2	8	72.7	1	9.1	0	0
	F+M	0	0	7	26.9	16	61.5	3	11.5	0	0
19–22	F	0	0	0	0	14	53.8	7	26.9	5	19.2
	M	0	0	0	0	14	93.3	1	6.6	0	0
	F+M	0	0	0	0	28	68.3	8	19.5	5	12.2
23–26	F	0	0	0	0	17	56.6	10	33.3	3	10
	M	0	0	0	0	4	44.4	2	22.2	3	33.3
	F+M	0	0	0	0	21	53.8	12	30.7	6	15.4
27–30	F	0	0	0	0	4	20	10	50	6	30
	M	0	0	0	0	4	25	6	37.5	6	37.5
	F+M	0	0	0	0	8	22.2	16	44.4	12	33.3

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
