# Peer review of "Analysis of the Frequency and Correlated Factors of Midpalatal Suture Maturation Stages in Young Adults, Based on Cone Beam Computed Tomography Imaging"

_jcm, 2022, doi:10.3390/jcm11236959_

Round 1

Reviewer 1 Report (Previous Reviewer 2)

Thanks very much for this interesting version of your paper. It has improved a lot from the previous version.

I want to highlight that Reference number 9 was published in 2022 and not 2021. Please correct the year of publication in the second paragraph in the Discussion section.

Also, please, in your Discussion, add this very relevant trial using CBCT images showing that 12- to 15-year-old patients could achieve very similar skeletal results in maxillary expansion when treated by slow or rapid maxillary expansion protocols:

Rabah N, Al-Ibrahim HM, Hajeer MY, Ajaj MA. Evaluation of rapid versus slow maxillary expansion in early adolescent patients with skeletal maxillary constriction using cone-beam computed tomography: A short-term follow-up randomized controlled trial. Dent Med Probl. 2022 Sep 15. DOI: 10.17219/dmp/133513. Epub ahead of print. PMID: 36108265.

Author Response

Good morning Dr:

Perfect, I have added that reference. Thank you very much!

Reviewer 2 Report (New Reviewer)

The authors present an analysis of the frequency and correlated factors of midpalatal suture maturation stages in young adults with maxillary constriction. The paper is easy to read and well written. My comments are listed below: 

1.     The title is too long and the abbreviation should not appear in it. Please change the title.

2.     The abbreviation should not also appear in the abstract or at least the full name should be given and then the abbreviation. Also remove the reference (14) from the summary!!! In two following sentences “In addition” is repeated, please delete one.

3.     Please change “Treacher Collins disease” into Treacher Collins syndrome!

4.     After introduction section the aim of the paper is missing, it is only present in the summary.

5.     There is a drawing of midpalatal suture morphology according to Angelieri et al but there is no subtitle of the figure and it is not cited in the text.

6.     In the sentence “The sample distribution showed 35.9% of male gender patients, with a total number of 51 subjects; while the female gender represented 64.1%, with 91 subjects” please remove the word “gender”.

7.     The discussion at the end is quite chaotic, especially when the paper of Reis et al is cited, I got confused which paragraph describes Reis article and which the Authors’ work, please rearrange it.

8.     Could you please extend your conclusions, maybe results obtained in your study could indicate, for example, the proper time of planned surgery? What is the take home message from your study?

Author Response

Good morning Dr:

Perfect, I have tried to correct everything mentioned. Thank you very much!

Round 2

Reviewer 2 Report (New Reviewer)

All remarks were taken into account. 

This manuscript is a resubmission of an earlier submission. The following is a list of the peer review reports and author responses from that submission.

Round 1

Reviewer 1 Report

The manuscript titled ”Evaluation of the frequency of the different stages of maturation of the midpalatal suture in a sample of young adults between fifteen and thirty years of age.” is interesting, but a serious revision is needed.

First of all, I appreciate the concern to plan a therapy based on reliable and not presumptive diagnostic data, especially regarding a traumatic therapy with heavy forces as rapid expansion of the maxilla (ERM) and consequent risks of important side effects (doi: 10.3390/children8050362). Unfortunately this concern is very rare among orthodontists.

  On the other hand, a logic and coherent reasoning from diagnosis to therapy is needed: when the lack of transversal development of the maxilla is of skeletal origin there is hypoplasia of the upper maxilla, mainly due to the lack of tensional forces from the masticatory function that are not enough to stimulate the bone apposition at the edges of the palatal suture which is a joint. This is the etiopathogenesis of the pathology and this is what orthodontists should correct. Breaking the suture means to take an immediate little advantage of less than half of the expansion realized by the screw (<4,5 cm) (doi: 10.1111/ocr.12244) with a great unavoidable relapse after 6 months and much more during time especially in growing subjects in which the suture will disappear and will not add bone any more. In adults, breaking the suture means to lose its viscoelasticity for ever. (ISBN 978-3-8055-8326-8)

INTRODUCTION

- Please the clinical description of the lack of transversal development shortening the introduction and add a clear description of the histological features of the palatal sutures both during growing and aging.

- To determine the degree of the bone maturity based on the morphology of the cervical vertebrae is no more considered the most reliable approach by the same authors who proposed it, please correct.

Please avoid the sentence “In the case of a skeletal origin, the appropriate treatment would be to perform a rapid expansion of the maxilla (ERM) or disjunction, according to which, based on the forces applied, we seek to separate the bony bases of the maxilla transversely, with the aim of making up for this lack of development by means of bone neoformation.” What we obtain with the heavy forces of the ERM is the destruction of the vascularity of the suture and apposition of scar tissue-like repair bone. The suture will disappear and will never be able to affix physiological bone or perform its very important viscoelastic action after ERM. A logical and consistent therapy with the diagnosis of lack of transversal development must stimulate the suture to produce physiological bone, not destroy it irreversibly.

MATERIALS AND METHODS

Subjects: being the ERM most used in children, even very little children(!), two groups under 15 years of age would have been important. Please explain why you did not consider these age groups.

DISCUSSION

It is true that “there is still no consensus protocol on the age limit for disjunction”, but it is not true that “This is mostly due to the enormous physiological variability of the mesopalatine suture between patients in terms of its level of obliteration.” The problem is that the physiology of the palatal suture is completely disregarded and not respected by ERM. There are physiological therapies able to correct the malocclusions together with the functions of the mouth in a stable way, in the respect of the physiology of the system. This is very important for the quality of life of any of us.

CONCLUSIONS

Please avoid “…and are susceptible to disjunction without surgical assistance”. Simply report the results.

Please avoid “Palatal disjunction in adults is a treatment that can be perfectly developed, as long as it presents an unconsolidated mesopalatine suture” We do not know the histological features in humans enough to be sure that a traumatic therapy as ERM will not have consequences for the life. Please correct accordingly.

Author Response

I upload a new article with all the corrections made.

Reviewer 2 Report

Thanks for this interesting paper.

However, I have the following points that require addressing:

Title

1- The title does not reflect the content of the paper. The authors should have included the following elements in the title: (1) The method used for evaluation, i.e., the CBCT imaging, (2) The descriptive and analytical nature of the study since the authors evaluated the distribution of the different categories of the classification, but they also analyzed the relationship between the developmental stage of the suture and age/sex, and (3) The study was confined to patients with maxillary constriction, i.e. the assessment was not performed on different types of skeletal malocclusion.

Abstract

2- The introduction of this abstract is written in a non-scientific manner. The authors are asked to write their statements with accurate words and phrases. The phrase "more and more" is not from the scientific writing of medical papers.

3- The aims should include the primary and secondary objectives. The authors did not mention the aim of analyzing the possible correlation between sex and maturation and age group and maturation.

4- In the methods section, the authors should provide more information about the method of classification used in their assessment of the collected CBCT images. Also, the study design is not given. General sentences about the statistical analysis employed are not given.

5- Results: Very limited. More results should be given here.

6- Conclusions: Age groups are not given in the conclusions section. In other words, the word "respectively" does not have related groups. Please correct. 

Introduction

7- In the first paragraph, please omit the word "etc." since this is not from formal scientific writing.

8- I was astonished to see all of the given text in the introduction about things that are not related to the actual content of this paper. The paper is about the different stages of maturation concerning different age groups of patients requiring maxillary expansion. Here, the authors produced several paragraphs about the transverse problems in orthodontics. This was followed by clinical and radiographic examination of patients with maxillary constriction and how to differentiate dental from skeletal problems. Many paragraphs appeared irrelevant to the actual topic of this manuscript. Therefore, I suggest minimizing all of these paragraphs or even deleting them.

9- Page 2, Line 67, the criticism of the landmarks used in the transverse assessment of the skeletal components of the face is not correct. I think this criticism should be modified and written in a better form.

10- Page 2, Line 73, the authors mentioned Reference 5 to support the benefits of using CBCT in assessing the midpalatal suture. Unfortunately, the Reference used is inappropriate and irrelevant. 

11- Page 2, Line 78, the authors should keep in mind that patients with maxillary deficiency at young ages (i.e., between 9 and 13) could benefit from the slow maxillary expansion that can expand the maxilla and open the midpalatal suture. Therefore, the statement should be adjusted for slow and maxillary expansion methodologies. But again, a sentence could be added that it is very common to resort to RME in adolescents to expand their constricted maxillae.

12- The well-known abbreviation for the rapid maxillary expansion is RME and not "ERM". Please use the current abbreviation in all the different areas of this manuscript. Please convert "ERM" to "RME" everywhere.

13- Page 2, Line 81, please replace the word "TORQUE" with a better concept which is "INCLINATION". Patients with dentally-based posterior crossbites require treatment with expansion and moving the crowns towards the vestibule with a change in the vestibulo-palatal inclination. The word "torque" could confuse the reader about the actual movement required. 

14- Actually, the beginning of the Introduction section should start from Page 2, Line 83. At this point, the orientation of the flow of paragraphs started to be toward midpalatal suture opening. This is the core of the introduction to justify the onset of this research.

15- Page 3, Line 89, the authors say that the method chosen to determine whether it is possible to perform RME or not came from the hand and wrist radiograph. This statement is incorrect and not supported by any relevant reference. Please rephrase it.

16- Page 3, Line 91, what is meant by the word "terradiography"? 

17- Page 3, Line 98, what is meant by the phrase "the occlusal plate"? Do you want to say "occlusal radiographs"?

18- The use of 'we' in different places in this manuscript is not favorable. The passive voice should be used and not the active voice. This problem appeared in different sections of this manuscript. Please correct all.

19- The shortcoming of using occlusal radiographs should be supported by a good reference and not Reference no 13. You need to find a paper discussing the diagnostic accuracy of occlusal radiographs in assessing the status of maturation of the midpalatal suture.

20- There is no need to repeat the definition of the different categories of the Classification of Angelieri et al. (2013). You just need to give a referral to their work. Also, you can explain what you need to explain in the given figure. Please omit unneeded sentences (i.e., from Line 107 to 119).

Materials and Methods

21- How did you explain the research purposes to patients if the study is retrospective? In other words, you do not need to obtain patients' consent to use their CBCT images already available at your department. 

22- Table 1 would be omitted and converted into text.

23- If the patient was18 years and six months, to which group did this patient belong? This should be given clearly when speaking about the age groups.

24- Page 4, Line 145, and Line 148, please give the whole details about the product (Product name, company name, city, country).

25- Page 4, Line 149, what do you mean by "a standard manner"? Please explain clearly the method of head orientation and positioning.

26- Please, the caption of Figure 01 should appear immediately beneath it and not at a distant place.

27- The whole section (Materials and Methods section) requires the use of some additional subheadings. Lengthy paragraphs are written without using proper subheadings. Please consider seriously adding additional subheadings to make the reading easy and smooth.

28- Page 5, Line 164, the format of giving three subsequent references should be written like this "(14-16)". Please double-check the guidelines regarding in-text citation. You need to conform to these guidelines.

29- Page 5, Line 177, the authors are asked to provide full bibliographic information regarding the mentioned source of the program used (i.e. the calculator).

30- Page 5, Line 182, how did the patients sign the consent forms of the study was retrospective?

31- Inclusion and exclusion criteria for the CBCT records should be given at the beginning or the middle of this section and not at the end.

32- Page 6, Line 195, what do you mean by this phrase "the maturation stage as a qualitative variable"? You need to be more specific. Please explain the dependent and independent variables in your logistic regression models.

33- According to my knowledge, the odds ratio is not a measure of association. Please consider revising this statement.

Results

34- First, using personal pronouns (we) is not a good way of scientific writing. You should use the passive voice when giving the results of the current research. Please consider visiting all the different paragraphs of this section.

35- A referral should be given to the appropriate table when giving the results of statistical testing. We need to say referrals to Table 1 or Table 2.  

36- The results are written with some interpretations of them. In this section, there is no place for giving explanations, showing emotions, or trying to connect results. Interpretations and discussion should be left to the appropriate place (i.e., the Discussion section).

37- Page 6, Line 227, the phrase "In general terms" should not appear in the Results section. You need to give the results without any general comments. Leave your comments in the Discussion section.

38- On page 6, Lines 232 to 234, the authors forgot that they had been writing the Results section and not the Discussion section!

39- No tables are given showing the results of the inferential statistics. These inferential statistics should be given in good tables. I am astonished not to see any table showing the results of the logistic regression performed.

40- The caption of Figure 2 does not correspond to the given legend. I wonder if the legend was already discussed somewhere in the main text. 

Discussion

41- Page 8, Line 258, the phrase "as I have mentioned above" is not a good way of scientific writing. The paper is not a chapter of a textbook. All phrases should be mentioned concisely, and remember that you are not writing a chapter or a textbook. Personal opinions should not appear anywhere in the main text.  

42- Using personal pronouns (we, they, you) is not a good way of scientific writing. You should use the passive voice when discussing the results of the current research. Please consider visiting all the different paragraphs of this section.

43- Page 8, Line 261, please explain these two abbreviations (MARPE, SARPE) at their first appearance giving attention to providing the relevant references.

44- When discussing the notion that the midpalatal suture can vary in the maturation stage among different age groups and there is no solid rule that RME cannot be applied at specific age times, you can give a citation to a recent study conducted on patients between 12 and 15 years with maxillary skeletal constriction (by Rabah et al., 2022) in which slow maxillary expansion was compared with rapid palatal expansion assuming the possibility to disjunction the midpalatal suture even when using slow expansion protocols.

((Here is the citation: Rabah N, Al-Ibrahim HM, Hajeer MY, Ajaj MA, Mahmoud G. Assessment of Patient-Centered Outcomes When Treating Maxillary Constriction Using a Slow RemovableVersus a Rapid Fixed Expansion Appliance in the Adolescence Period: A Randomized Controlled Trial. Cureus. 2022 Mar 3;14(3):e22793. doi: 10.7759/cureus.22793.PMID: 35261839; PMCID: PMC8893008.))

Conclusions

45- The use of the word 'respectively' was given without mentioning the age group to which you are referring the reader. Please add the age groups being studied. 

46- Please use the passive voice in these sentences. Regarding the second conclusion, you need to use proportions instead of "high numbers of individuals."

47- Is the third conclusion supported by the given results in this work?

References

 48- Reference 10 required additional information.

49- References 13 and 14 are duplicate sources. Please delete 14 and re-number your references again.

50- The whole references require attention since they are not formulated using the same style. I wonder if they conform to the style of this journal. For example, in Reference 17, The number of the volume appeared first. Then this was followed by the Journal title. At the end of the citation, the year of publication appeared (without the issue number), then the pages were mentioned using the letter 'p'. 

Author Response

(The authors gave the same response as above.)
